# Complementary Foods and Milk-Based Formulas Provide Excess Protein but Suboptimal Key Micronutrients and Essential Fatty Acids in the Intakes of Infants and Toddlers from Urban Settings in Malaysia

**DOI:** 10.3390/nu13072354

**Published:** 2021-07-09

**Authors:** Geok Lin Khor, Siew Siew Lee

**Affiliations:** 1Department of Nutrition and Dietetics, International Medical University, Kuala Lumpur 57000, Malaysia; 2Department of Nutrition and Dietetics, Universiti Putra Malaysia, Serdang 43400, Malaysia; siewsiew_lee89@hotmail.com

**Keywords:** complementary foods, milk-based formulas, energy, protein, micronutrients, fatty acids intake

## Abstract

This study determined the intakes of complementary foods (CFs) and milk-based formulas (MFs) by a total of 119 subjects aged 6–23.9 months from urban day care centers. Dietary intakes were assessed using two-day weighed food records. Intake adequacy of energy and nutrients was compared to the Recommended Nutrient Intakes (RNI) for Malaysia. The most commonly consumed CFs were cereals (rice, noodles, bread). The subjects derived approximately half of their energy requirements (kcals) from CFs (57 ± 35%) and MFs (56 ± 31%). Protein intake was in excess of their RNI requirements, from both CFs (145 ± 72%) and MFs (133 ± 88%). Main sources of protein included meat, dairy products, and western fast food. Intake of CFs provided less than the RNI requirements for vitamin A, thiamine, riboflavin, folate, vitamin C, calcium, iron, and zinc. Neither CF nor MF intake met the Adequate Intake (AI) requirements for essential fatty acids. These findings indicate imbalances in the dietary intake of the subjects that may have adverse health implications, including increased risk of rapid weight gain from excess protein intake, and linear growth faltering and intellectual impairment from multiple micronutrient deficiencies. Interventions are needed to improve child feeding knowledge and practices among parents and child care providers.

## 1. Introduction

Globally, malnutrition continues to exert a heavy toll on young children. Almost 200 million children under five suffered from stunting or wasting in 2018 [1]. Undernutrition not only results in poor physical growth and development but is also known to have an adverse impact on cognitive, motor, language, and socio-emotional skills [2]. Prolonged undernutrition leads to compounded risks of morbidity and mortality [3].

Meanwhile, childhood obesity, the flip side of the same “malnutrition coin” continues to escalate. In 2019, an estimated 38.2 million children under the age of five years were overweight or obese. Once considered a high-income country problem, overweight and obesity are now on the rise in low- and middle-income countries, particularly in urban settings. Almost half of the children under five who were overweight or obese in 2019 lived in Asia [4].

Worldwide, the pathogenesis of childhood malnutrition has been extensively investigated—in Africa [5,6,7], South America [8,9], and in Asia [10,11,12]. Despite the geographical and cultural diversity of the communities studied, these studies reported several common factors that are associated with the plight of undernourished children. These include low socio-economic status, household food insecurity, unhealthy living environment, low maternal nutritional status, suboptimal breastfeeding, and complementary feeding practices. 

There are comparatively fewer studies that quantitatively assessed the adequacy of energy and nutrient intake as proximal determinants of childhood malnutrition. Based on 24-h recalls collected of infants and toddlers aged 9–24 months in selected communities in South America, Africa, and Asia, Maciel et al. (2021) [13] reported that higher intakes of energy and zinc in complementary feeding were associated with decreased risk of malnutrition. In a study of infants at ages 2, 5, and 12 months from low-resource Indonesian households, Leong et al. (2021) [14] conducted an assessment of intakes from both human milk and complementary foods. Their findings included inadequate intakes of iron, thiamine, niacin, and vitamin B12 by a high prevalence of the infants, with manifestation of growth faltering during the critical 6 months of early infancy. In Malaysia, Khor et al. (2016) [15] reported inadequate dietary intake of infants and toddlers that failed to meet the daily recommendations for several micronutrients, including vitamin A, thiamine, vitamin C, niacin, riboflavin, and zinc. 

This manuscript is a follow-up of the preceding study published by Khor et al. (2016) [15]. In this report, we aimed to quantitatively determine the intake of energy, key macronutrients, and micronutrients from (i) complementary foods and (ii) milk-based formulas separately, using two-day weighed food records among infants and toddlers aged 6–23 months living in the urban suburbs of Kuala Lumpur and Putrajaya, Malaysia.

## 2. Materials and Methods 

### 2.1. Study Design and Subjects

Details of the study design have been described in a previous publication [15] that focused on compliance with WHO Infant and young child feeding (IYCF) [16] among Malaysian infants and toddlers. A cross-sectional study design was used to recruit the computed sample size of 300 infants and toddlers from licensed child care centers (Licensed child care centers are registered with the government Social Welfare Department, and are obliged to adhere to government regulations including caregiver to child ratio, standards of hygiene practice, and serving meals in compliance with menu guidelines from the Ministry of Health) located in the suburbs of Kuala Lumpur and Putrajaya. Names and addresses of licensed child care centers were obtained from the Association of Registered Care Providers of Selangor State and the Ministry of Health Nutrition Division. The child care centers were contacted through emails and phone calls. Out of approximately 100 centers contacted, one-fifth gave consent to participate in the study. Using a convenience sampling approach in going from one center to the next, approximately 100 infants and children who met the inclusion criteria were recruited for each of these age groups: 6.0–11.9 months, 12.0–17.9 months, and 18.0–23.9 months. The following were the study’s inclusion criteria: (i) aged 6.0 to 23.9 months; (ii) taking meals at the selected child care centers, and (iii) of Malay ethnicity. The age disaggregation was in accordance with the recommendation of WHO (2010) for IYCF studies. Each of the age groups consisted of approximately an equal proportion of male and female subjects. Only Malay subjects from urban areas were selected in order to minimize the influence of cultural and socio-economic factors on child feeding practices. Meanwhile, infants with mental or physical disability, those who were ill at the time of data collection, or those who had dietary restrictions were excluded from the study. As the participants were recruited by convenience sampling, they were not deemed as representative of urban Malay toddlers. 

For the quantitative estimation of dietary intake reported in this manuscript, 40 subjects, out of the original 100 subjects in each age group, were selected based on convenience sampling. Written consent was sought from parents to participate in the dietary intake part of the study. When consent was obtained, his/her child was slotted into the appropriate age/gender group (6.0–11.9 months, 12.0–17.9 months, or 18.0–23.9 months). In this way, a total of 120 subjects was obtained. Each age group consisted of approximately an equal number of male and female subjects. At the end of the study, one dietary intake questionnaire was found incomplete and, thus, only 119 records were available for data analysis. 

### 2.2. Estimation of Dietary Intake 

Senior nutrition undergraduate students from the Department of Nutrition and Dietetics, International Medical University, were employed as research assistants for estimation of dietary intake in this study. They had received practical training on dietary intake procedures including weighing intake methods in their academic program. For this study, they received further practical demonstrations in order to standardize the procedure and minimize errors. They carried out the weighing of intake of food and beverages over two consecutive weekdays when the child was in the center from about 7 a.m. until 5 p.m. Food items were weighed using a kitchen scale (TANITA-KD160WH) that has a weight capacity of 2 kg and a weight graduation of 1 gm. The food and beverages consumed were computed by deducing the amounts not consumed by the child (leftovers) from the total amounts weighed before they consumed. Detailed recipes for food consumed at the centers were obtained from the center manager or person who prepared the foods and beverages. 

For recording food and beverages consumed by subjects who were fed individually by the center helpers, the student research assistants first identified the ingredients used in preparing the food or beverage, and then estimated the amounts of the food/beverage consumed by the child, based on the serving spoon or cup. As the entire procedure of observation, weighing, and calculation for each child was time-consuming, a research assistant could complete the procedure for only three to four subjects a day, on average.

All parents were provided with a simple template and instructed to record foods and beverages, in terms of description and quantity, taken by their child between the time the child left the center in the evening until the next morning when the child was brought back to the center. The research assistants checked the forms and sought clarifications from parents, particularly for the quantities and types of constituents in the food or beverage consumed at home or outside. 

### 2.3. Determination of Energy (kcals) and Nutrient Contents

The Malaysian Food Composition database [17] was the primary source for obtaining the contents of energy and nutrients of the foods and beverages consumed by the subjects. Energy intake was computed directly from the food intake in kcal. Besides energy, protein; total fat; and the micronutrients vitamin A, thiamine, riboflavin, niacin, vitamin C, calcium, iron, and zinc were obtained from the Malaysian food composition database. The Singapore Energy and Nutrient Composition of Food Database [18] and the United States Department of Agriculture National Nutrient Database [19] were also used as reference for contents of folate, fats (PUFA, MUFA, SFA), and fatty acids. 

Food matching was performed according to the guidelines of the Food and Agriculture Organization (FAO) and international Network of Food Data Systems (INFOODS) [20]. In order to correct for potential differences in the amount of total fat in US and Malaysian foods, as stated in the Malaysian Food Composition Table [17], imputed nutrient values of the matched food database were adjusted with a factor I = total fat Malaysia/Fat USDA. As for packaged food and commercial food such as infant formula and baby foods, information on energy and nutrient content were obtained from the companies’ product labels or websites. Fatty acids, vitamin D, and zinc contents were also obtained from published articles that had reported analytical results of these nutrients in some Malaysian foods [21,22]. 

Food and nutrient intake per day were reported as mean ± standard deviation (SD) and median based on the average of the two-day weighed food records. Intake of energy and nutrients from complementary food items were computed separately from those taken from formula milk. The adequacy of nutrient intake was determined by comparing the individual total daily intake with the age-appropriate Recommended Nutrient Intakes (RNI) for Malaysia [23]. 

### 2.4. Anthropometric Measurements

Body weight and linear length of the subjects were measured using approved methods and validated equipment [24]. The means of duplicated measurements for each subject were computed and compared against the 2006 WHO Child Growth Standards for children below 5 years [24]. Children with length-for-age z-score <−2 were categorized as stunted, weight-for-age z-score <−2 as underweight, and >+2 z-score as overweight, while weight-for-height z-score <−2 as wasted. 

### 2.5. Statistical Analysis

All statistical analyses were performed using IBM SPSS version 21.0 (IBM Corp., Armonk, NY, USA). Characteristics of the infants and toddlers were described as frequencies for categorical variables. Continuous variables were described as mean, standard deviation (SD), and median. One-way ANOVA was used to compare the continuous variable between aged group, while Chi-square and Fisher’s Exact tests were used to compare the categorical variables between age group. Kruskal–Wallis test and Dunn–Bonferroni post hoc were used to test the statistical difference between three age groups for the selected nutrients and food group intake. Statistical significance was set at *p* < 0.05. 

This study was approved by the Joint Committee on Research and Ethics of the International Medical University, Kuala Lumpur, Malaysia (Project ID no. IMU R 123/2013). Informed consent to participate in the study was obtained from the parents prior to data collection. 

## 3. Results

### 3.1. Demographic, Feeding, and Anthropometric Background of Subjects

In this cross-sectional study, a total of 119 subjects were included, out of whom, 34 were aged 6.0–11.9 months, 41 were in the 12–17.9 months category, while another 44 were of ages 18–23.9 months (Table 1). Overall, there were slightly more females than males among the younger subjects, while for the oldest age category, there were equal proportions of male and female subjects.

Mothers were asked if their child was breastfed or formula-fed or both on the day before the interview day. There were significantly more of the youngest subjects (6–11.9 months) who were breastfed (23.5%) than among the oldest (18–23.9 months) (6.8%). In contrast, a significantly higher percentage of the oldest subjects (63.6%) were formula-fed compared to the youngest (20.6%). Nonetheless, it is notable that a significantly higher prevalence of the mothers of the youngest age group was practicing mixed feeding, that is, breastfeeding and formula-feeding (55.9%) compared to 29.5% among mothers of the oldest age group. 

Based on the anthropometric measurements, 23.5% of the subjects were classified as stunted (length-for-age z-score <−2SD), while 9.2% were underweight (weight-for-age z-score <−2SD) and another 8.4% were wasted (weight-for length z-score <−2SD). The percentage of undernutrition according to these indicators did not differ significantly across the age categories. It is noted that, overall, 11.4% (*n* = 7) of the subjects were both stunted and underweight, while another subject (*n* = 1) was both stunted and wasted these manifestations of the double burden of undernutrition among the toddlers, albeit affecting a small number, is a noteworthy observation. Meanwhile, overweight appears not to be a concern in this cohort of subjects, as only one of the older toddlers was determined as overweight. 

### 3.2. Intake of Complementary Foods

#### 3.2.1. Estimation of Quantity of Complementary Foods Consumed According to Age Group

The types and mean quantity of complementary foods (Complementary foods are foods or drinks other than breast milk or infant formula (liquids, semisolids, and solids) introduced to an infant to provide nutrients. (USDA) https://wicworks.fns.usda.gov/wicworks/Topics/FG/ accessed 27 January 2021) consumed by the subjects per day, as determined using two-day weighed food records, are tabulated in Table 2. Overall, cereals constituted the most commonly consumed type of complementary food, taken by almost 100% of the subjects on a daily basis during the study. Cereals are consumed as rice, porridge, noodles, bread, and breakfast cereals. Overall, the mean quantity of cereals taken per day was 179.5 ± 85.4 g, with significantly higher intakes by the older age groups (204.7 ± 78.8 g by 12–17.9 months and 198.9 ± 63.4 g by 18–23.9 months) than that taken by the youngest group (123.9 ± 94.1 g by 6–11.9 months). 

Milk, including ultra-high temperature (UHT) milk, fresh milk, flavored milk (chocolate milk), and dairy products (yogurt, cheese) formed the next highest quantity of complementary food consumed per day by the two older age groups, 45.5% and 24.4% among the 18–23.9 months and 12–17.9 months, respectively. Other major protein complementary foods, namely, chicken, eggs, and fish were also consumed in significantly higher quantities by the older groups compared to the youngest subjects. Chicken and fish were usually cooked in porridge. The older subjects also derived protein from “western foods” such as nuggets, frankfurters, and hot-dogs.

Meanwhile, fruit and vegetable consumption in terms of mean intake per day showed no statistical difference across the age groups. However, the forms of fruit and vegetables eaten differed by age group. While the older subjects consumed vegetables prepared as soup, purees, and cooked vegetables, the youngest group was fed commercially prepared baby fruit foods and baby vegetable foods. Other types of commercial baby foods consumed by the youngest group were baby cereals and teething biscuits. 

#### 3.2.2. Energy (kcals) and Nutrient Intake from Complementary Foods According to Age Group

Based on the types and quantity of complementary foods consumed described above, the quantity of energy (kcals) and nutrients (g) consumed were computed for each subject. Table 3 shows the intakes recorded as mean ± SD and median for each age group. In addition, the intakes of energy and nutrients were compared to the recommended daily intake (RNI) for Malaysia appropriate for the ages of the subjects. 

The mean intake of energy per day by 6–11.9 months was 415.0 ± 308.2 kcals, whilst that of the 12–17.9 months and 18–23.9 months were significantly higher at 510.0 ± 307.8 kcal and 543.3 ± 241.1 kcal, respectively. These levels of energy intake, on average, met 58 ± 34.6% of the RNI, there being no significant difference among the age groups. It is noted that the SD for energy intake within each age group was wide, indicating notable variations in the intake levels among the subjects belonging to the same age group. 

As for protein intake, the result showed significant differences among the age groups, with the youngest consuming, on average, 13.3 ± 8.6 g per day, compared to 17.3 ± 8.2 g and 19.3 ± 8.0 g, respectively, for ages 12–17.9 months and 18–23.9 months. At these intake amounts, consumption of protein from complementary foods exceeded the RNI levels for the respective age groups. Overall, for all the age groups, the average protein intake derived from complementary foods exceeded the daily recommendation at 147.1 ± 73.3%, with the oldest subjects (18–23.9 months) consuming significantly higher intakes at 160.6 ± 66.7%. 

Besides protein, the intake adequacy of other nutrients from complementary foods that were computed included vitamin A, thiamine, riboflavin, niacin, folate, vitamin D, vitamin C, iron, calcium, and zinc, as suggested by WHO (2009) (Good complementary foods are rich in energy, protein, and micronutrients, particularly iron, zinc, calcium, vitamin A, vitamin C, and folate). Overall, for all the age groups, the mean intakes of all these nutrients in question were insufficient as none achieved their respective RNI levels (Table 3). Out of these, the overall mean intakes of vitamin A, thiamine, riboflavin, niacin, and iron met between 50–90% of their RNI levels, while that of vitamin D, calcium, zinc, and folate attained less than 10–35% of their respective RNI values. 

Vitamin D intake was the lowest in meeting only 11.1 ± 16.8% of the RNI for 6–11.9 months, 4.8 ± 6.2% for 12–17.9 months, and 7.0 ± 9.4% for 18–23.9 months. The next deficient nutrient is calcium as its intake was noticeably inadequate, especially among the older subjects. On average, calcium consumption from complementary foods attained the low levels of 15.2% and 23.8% of the RNI for 12–17.9 months and 18–23.9 months, respectively. In comparison, the mean intakes of B vitamins, namely, thiamine, riboflavin, and niacin, were noticeably higher, meeting 67.1%, 88.3%, and 73.1%, respectively, of the RNI levels for the overall ages, although their intake also did not reach the RNI levels for any of the age groups. The results showed that iron intake from complementary foods was relatively adequate in attaining 72.8% of the RNI level overall. In contrast, zinc intake met only 35.1% of the overall RNI for all ages. 

In summary, with the exception of protein, complementary foods consumed by the study subjects provided inadequate energy and several key nutrients, especially vitamin D, calcium, zinc, and folate. 

#### 3.2.3. Fats and Fatty Acids Intake from Complementary Foods According to Age Group

Intake of fats and fatty acids were analyzed separately to determine adequacy intake of the essential fatty acids, namely, linoleic acid (LA n-6 PUFA) and alpha linolenic acid (ALA N-3 PUFA). Intake of fats and fatty acids per day by the subjects are shown in Table 4. Based on the 30–40% total energy intake (TEI) recommended as fat calories for infants and toddlers in the 2017 RNI for Malaysia (630 kcal for boys and 570 kcals for girls aged 6–8 months, 720 kcals for boys and 660 kcals for girls aged 9–11 months (FAO, 2010; NCCFN, 2017)), the RNI values for total fat for the subjects were computed accordingly (Boys: 6–8 months, 21–28 g/day; 9–11 months, 24–32 g/day; 1–3 years, 27–38 g/day. Girls: 6–8 months, 19–25 g/day; 9–11 months, 22–29 g/day; 1–3 years, 25–35 g/day).

The amounts of total fat from complementary foods consumed by all the age groups were low in terms of meeting their recommended intake levels. For ages 6–11.9 months, total fat intake was 3.5 g/day, contributing to 14.1 ± 10.6% of RNI, while that for ages 12–17.9 months and 18–23.9 months were 29.7 ± 20.3% and 38.8 ± 20.3%, respectively. As for the essential fatty acids, the intake of LA and ALA were compared to the Adequate Intake (AI) values recommended by US Institute of Medicine (IOM, 2002). AI values are assigned in view of the small intakes of LA and ALA, and a recommended dietary allowance (RDA) could not be determined (Devaney and Barr, 2002). In this study, the subjects of all ages were shown to consume inadequate levels of both the essential fatty acids from complementary foods. For the youngest to the oldest age groups, their intake of LA ranged from 15.7 ± 15.2% to 26.3 ± 16.0%, while that for ALA ranged from 13.7 ± 14.5% to 24.6 ± 17.4% of their respective AI values (Table 4).

### 3.3. Intake of Commercial Infant/Child Milk-Based Formulas 

#### 3.3.1. Estimation of Quantity of Commercial Milk-Based Formulas Consumed According to Age Group 

A total of 15 different brands of commercial milk-based formulas were taken by the subjects during the study period. Out of these, 12 were cow milk-based formulas, 2 were goat milk-based, and 1 was soymilk-based formula (Table 5). Almost three-quarters of the subjects (74.8%) were taking only five of the 12 brands of cow milk-based formulas. Thus, while many brands of cow milk-based formulas are available commercially, only a few brands are consumed by the majority of the infants and toddlers in this study.

The most popular brand is coded here as Formula C1 and it has three categories according to its targeted age group, namely, C1 Step 1 Infant Formula Milk (0–12 months), C1 Follow-up Formula Milk (6–18 months), and C1 Step 3 Plain Formulated Milk (1 year and above). All the rest of the milk-based formulas, including the goat milk-based formulas, also have different categories targeted at different age groups. It is evident that a single brand of milk-based formula comprises various categories to cater for a wide age range of infants and toddlers.

It is noted that not all subjects in the youngest group (6–11.9 months) were consuming formulas, but reportedly only 79.4% of them were (Table 5). By comparison, as high a percentage of the older subjects were consuming formulas, that is, 75.6% among ages 12–17.9 months and 84.1% in the 18–23.9 month group. The mean quantity of all formulas consumed by ages 6–11.9 months was 78.8 ± 42.5 g per day, while that taken by the 12–17.9 months and 18–23.9 months were 115.0 ± 76.3 g and 112.3 ± 52.7 g per day, respectively. In brief, as high a prevalence of the older subjects were consuming formulas, and they were also consuming higher amounts of formulas per day than the infant group. 

#### 3.3.2. Energy (kcals) and Nutrient Intake from Commercial Milk-Based Formulas According to Age Group

Estimation of intake of energy (kcals) and nutrients from milk-based formulas is based on a total of 95 subjects only as not all the subjects reported consuming formulas. Overall, the mean intake of energy was 480.6 ± 276.1 kcals per day, which amounted to meeting 56.3 ± 31.0% of the RNI for the subjects overall (Table 6). Consumption of formulas provided about half of the RNI requirement for energy for each age group, ranging from 58.4 ± 31.6% for the youngest to 54.2 ± 24.9% for the oldest group. In this respect, there was no significant difference among the age groups. 

In contrast, protein intake from milk-based formulas provided more than the RNI requirement levels overall (133.1%). Formulas drinking provided 100.9 ± 64.1% of the RNI requirement for the youngest age group to a significantly higher level of 149 73.3% for the oldest subjects. As for fat intake, formulas provided about two-thirds of its RNI, being slightly higher for the youngest age group (72 ± 41.2%) and at 69 ± 45.7% and 60.5 ± 29%, respectively, for the ages of 12–17.9 months and 18–23.9 months. 

This study found that consuming milk-based formulas provided more than the RNI requirements for vitamin A, thiamine, riboflavin, folate, vitamin C, calcium, iron, and zinc, but did not meet the RNI levels for total fat, niacin, and vitamin D. It is noted, nonetheless, that the youngest age group was the worst off among the age groups in terms of not meeting their RNI requirements for energy, total fat, vitamin A, vitamin D, niacin, iron, and zinc from consuming milk-based formulas. This may be due to the relatively smaller amounts of formulas taken by the youngest group, estimated at about 78.8 g (Table 5) or the equivalent of 5 tablespoons/day, compared to about 8 tablespoons/day for the older subjects. 

#### 3.3.3. Fats and Fatty Acids Intake from Milk-Based Formulas According to Age Group

Milk-based formulas provided almost two-thirds of the RNI for total fat, at 72 ± 41.2% among the youngest (6–11.9 months), 69 ± 45.7% (12–17.9 months), and 60.5 ± 29% for the oldest (18–23.9 months) (Table 7). As for intakes of the essential fatty acids, milk-based formulas provided relatively higher amounts of both LA and the ALA compared to complementary foods. While previously noted that complementary foods provided about 16–26% of the AI levels for LA and 14–25% for ALA + DHA + EPA among the different age groups, milk-based formulas provided 52.8 ± 31.4% of the AI for LA for 6–11.9 months and 39.9 ± 26.7% for 12–17.9 months, and was somewhat lower at 33.3 ± 24.1% for the oldest age group. The formulas were as good a source of ALA in meeting the AI levels ranging from 44.2 ± 25.7% (18–23.9 months) to 47.9 ± 33% (12–17.9 months). Still, it should be noted that neither complementary foods nor milk-based formulas were providing enough total fat and essential fatty acids for the infants and toddlers under study.

### 3.4. Relative Proportions of Energy and Nutrient Intake from Complementary Foods and Commercial Milk-Based Formulas According to Age Group

This study also investigated the comparative intakes of energy and selected key nutrients from complementary foods and milk-based formulas amongst subjects who consumed both (*n* = 95) (Table 8). Overall, the subjects derived approximately half of their energy requirements (kcals) from complementary foods (58 ± 49% of RNI) and about half from milk-based formulas (56 ± 31%) per day during the study period. Taken together, complementary foods and milk-based formulas appeared to provide all the age groups with enough energy to meet their energy RNI levels. 

As for protein intake, the subjects consumed excess amounts of protein from both complementary foods (147 ± 73% of RNI) and milk-based formulas (133 ± 88% of RNI). For each age group, protein intake from complementary foods was higher than that derived from milk-based formulas. 

Complementary foods compared less well with milk-based formulas in meeting lower requirements levels of total fat, vitamin A, iron, zinc, calcium, and folate for each age group. Milk-based formulas provided a higher mean amount of total fat for ages 6–11.9 months, and also provided a higher RNI level of iron for ages 18.0–23.9 months. In summary, milk-based formulas were a more adequate source of total fat and several nutrients compared to the complementary foods taken by the subjects.

## 4. Discussion

As a child grows from birth through early childhood, lean body mass increases rapidly, and the energy required to synthesize tissues and for homeostasis of metabolically active tissues begins to exceed the calories provided from exclusive breastfeeding [25]. Without additional calories in the diet, a negative energy balance will develop that eventually leads to undernutrition [26]. Hence, ensuring calorie needs are met is WHO’s primary guiding principle for feeding infants [27]. Butte (2005) [28] recommended calorie intake from breast milk and complementary food at approximately 600 kcal per day at 6–8 months of age, 700 kcal per day at 9–11 months of age, and 900 kcal per day at 12–23 months of age. 

In this study of subjects aged 6–23.9 months, neither complementary foods nor milk-based formulas taken on their own provided sufficient calories to meet their RNI requirements. However, taken together, complementary food and milk formulas appear to meet their age RNI requirements for calories. The main complementary food sources of calories include commercial baby foods (cereals, biscuits), cereals (rice, bread, noodles, pasta), and western foods (burger, nuggets, pizza). 

Besides energy, the importance of macro- and micronutrients for the growth and development of infants is also well recognized. Breast milk alone does not provide sufficient quantities of several key nutrients, including iron, zinc, vitamin D, vitamin B12, folate, and long-chain fatty acids, especially for older infants [29]. As such, the timing of introduction, content, and nutrient bioavailability in complementary foods are very important considerations, particularly for exclusively breastfed infants [30]. 

This cohort of infants and toddlers were found to consume protein in excess of their RNI requirement levels, from either complementary foods or milk-based formulas. The main complementary food sources of protein include meat, fish, and western foods. Meanwhile, the top brands of milk-based infant formulas favored by the subjects were shown to provide approximately 1.96 to 2.10 g protein/100 kcal. These protein levels are deemed relatively higher than that in breast milk (1.0–1.3/100 kcals) [31,32]. This discrepancy has been considered the key contributor to the greater weight gain observed in formula-fed infants based on the “early protein hypothesis”, as reviewed by Luque et al. (2016) [33]. This hypothesis suggests that a higher protein intake from milk-based formulas increases certain circulating essential amino acids, which stimulate the secretion of insulin and insulin-like-growth factor 1 (IGF-1), resulting in rapid weight gain and body fat deposition. Randomized controlled trials have been conducted to test this hypothesis. In comparing iso-caloric infant formula with high-protein (2.9 g/100 kcal) and low-protein (1.7 g/100 kcal) contents from birth to 12 months, Koletzko et al. (2016) [34] showed that infants who consumed the low-protein formula had a slower growth trajectory, similar to breastfed infants. Martin et al. (2014) [35] also showed elevated risks of becoming obese at 2 years of age in children fed standard feeding formula (2.7 g/100 kcals) compared to those fed a lower protein alternative (1.65 g/100 kcals). In a prospective study to examine infant growth in lower volume formula milk feeding (LFM, <840 mL formula/d), and higher-volume formula milk feeding (HFM, ≥840 mL formula/d) among 1093 Chinese infants, Huang et al. (2018) [36] reported that feeding higher volumes of formulas in early infancy is associated with greater body weight and overweight in later infancy. A systematic review to assess the macronutrient and energy content plus volume of intake in breast-fed and formula-fed infants in early infancy concluded that formula-fed infants have a 1.2-to 9.5-fold higher energy intake and a 1.2-to 4.8-fold higher protein intake than those who are breastfed in the first week of life [37]. Thus, formula-fed infants consume a higher volume and more energy dense milk in early life, leading to faster growth, which could potentially program a greater risk of long-term obesity. 

The evidence described above were based mostly on the first year of life, whereas the child’s high-weight-gain velocity period extends up to about 2 years of age. Thus, Ferré et al. (2021) [38] conducted a systematic review to investigate whether protein intake during the second year of life is associated with higher weight gain and obesity risk later in childhood. They found moderate evidence for the association between protein intake during the second year of life and increased body fatness at 2 years. Nonetheless, evidence supporting either an increased risk of overweight or over fatness later in life was inconclusive.

In assessing the intake of several key micronutrients from complementary foods and milk-based formulas, this study found that complementary foods compared poorly with milk-based formulas in meeting significantly lower requirements levels of total fat, vitamin A, iron, zinc, calcium, and folate for each age group. This finding calls for nutrition and education interventions as the period from 6 to 24 months of age is well-recognized as the time of peak incidence of growth faltering and infectious illnesses [39,40]. The majority of interventions during this vulnerable period have been targeted at improving infant and young child feeding (IYCF) practices [41]. Interventions to improve IYCF have generally focused on nutrition counselling, providing complementary food with or without micronutrients, and increasing energy density of complementary foods through simple technology [3].

There are limitations of this study that should be considered when interpreting our results. As the quantitative intake of breast milk was not determined, our findings on the dietary intake of the subjects are restricted to consumption of complementary foods and milk-based formulas only. Hence, results with regard to meeting RNIs for energy, macronutrients, and micronutrients of the subjects should be interpreted with caution. The accuracy of the results may be affected by the methods of data collection, particularly the weighing of food and beverages consumed. While training was provided to the research assistants to ensure that standardized procedures were used consistently, we did not record quantitatively the extent of any errors that might have occurred among the assistants. Another likely source of errors could be the estimation of intake by parents. Use of various food composition tables as reference sources for energy and nutrient contents of foods and beverages consumed may be another source of errors. As this article focused only on the dietary intake of infants and toddlers, risk factors that may affect consumption behavior of these young children were not included. The dietary results obtained should not be extended to children who do not attend the same day care centers, as the feeding programs may be different. 

To the best of our knowledge, this is the first study to assess the dietary intake of infants and toddlers in Malaysia using two-day weighed food records. By this means, quantitative data were obtained that provided evidence of (i) excess intake of protein from both complementary foods and milk-based formulas, which has implications of increasing the risks of overweight and obesity in older childhood and later life; and (ii) suboptimal intake of key micronutrients and essential fatty acids, especially from complementary foods that could predispose young children to risks of multiple micronutrient deficiency. The serious consequences of multiple micronutrient deficiencies to poor growth, intellectual impairments, perinatal complications, and increased risk of morbidity and mortality have been well reviewed [42]. 

## 5. Conclusions

This study provides current information on the intakes of energy and nutrients from complementary foods and milk-based formulas among subjects aged 6–23.9 months from urban settings in Malaysia. The findings indicate imbalances in the dietary intake of the subjects that could have adverse implications on the health and nutritional status of the infants and toddlers. Appropriate strategies and interventions are needed to improve the current IYCF knowledge and practices among parents and childcare providers, including less reliance on commercial and processed foods and greater use of homemade complementary foods. It is important to ensure the young child consumes not only the right food but also enough of the right food. 

## Figures and Tables

**Table 1 nutrients-13-02354-t001:** Nutritional and feeding status of subjects.

Variables	Total (N = 119)	6–11.9 Months (*N* = 34)	12–17.9 Months (*N* = 41)	18–23.9 Months (*N* = 44)	*p*-Value ^1^
	Mean ± SD/ N (%)	Mean ± SD/ N (%)	Mean ± SD/ N (%)	Mean ± SD/ N (%)	
Infant age (months)	15 ± 5	9 ± 2 ^a^	15 ± 2 ^b^	20 ± 2 ^c^	0.0001
Infant Sex					
Male	53 (44.5)	12 (35.3)	19 (46.3)	22 (50.0)	0.414
Female	66 (55.5)	22 (64.7)	22 (53.7)	22 (50.0)	
Feeding practice ^2^					
Breastfed	22 (18.5)	8 (23.5) ^a^	11 (26.8) ^a^	3 (6.8) ^b^	0.001
Formula fed	54 (45.4)	7 (20.6) ^a^	19 (46.3) ^a,b^	28 (63.6) ^b^	
Mixed	43 (36.1)	19 (55.9) ^a^	11 (26.8) ^b^	13 (29.5) ^b^	

Length (cm)	75.1 ± 5.9	68.9 ± 4.2 ^a^	75.1 ± 3.6 ^b^	79.9 ± 4.1 ^c^	0.0001
Length for age, z-score	−1.1 ± 1.4	−1.1 ± 1.4	−1.0 ± 1.3	−1.1 ± 1.5	0.959
Stunted ^3^	28 (23.5)	6 (17.6)	9 (22.0)	13 (29.5)	0.318
Normal	91 (76.5)	28 (82.4)	32 (78.0)	31 (70.5)	

Weight (kg)	9.2 ± 1.4	8.1 ± 1.0 ^a^	9.1 ± 0.9 ^b^	10.2 ± 1.3 ^c^	0.0001
Weight for age, z-score	−0.8 ± 1.0	−0.6 ± 1.1	−0.9 ± 0.8	−0.8 ± 1.1	0.400
Underweight ^4^	11 (9.2)	2 (5.9)	4 (9.8)	5 (11.4)	0.785
Normal	107 (89.9)	32 (94.1)	37 (90.2)	38 (86.4)	
Overweight ^5^	1 (0.8)	0 (0.0)	0 (0.0)	1 (2.3)	

Weight for length, z-score	−0.3 ± 1.3	0.1 ± 1.2	−0.5 ± 1.2	−0.4 ± 1.3	0.098
Wasted ^6^	10 (8.4)	1 (2.9)	5 (12.2)	4 (9.1)	0.380
Normal	109 (91.6)	33 (97.1)	36 (87.8)	40 (90.9)	

Stunted and Underweight	7 (5.9)	1 (2.9)	1 (2.4)	5 (11.4)	0.622

Stunted and wasted	1 (0.8)	0 (0.0)	0 (0.0)	1 (2.3)	-

^1^ Statistical analysis was conducted for comparing variables among the age groups using ANOVA for continuous variables and Chi-square and Fisher’s Exact test for categorical variables; ^a–c^ values that do not share the same alphabets were significantly different from each other within the row at *p*-value < 0.05. ^2^ Feeding practice as reported by mother/care taker for the previous day prior to the survey day; mixed d type refers to breastfed as well as formula fed. ^3^ Stunted, height/length for age z-score <−2 SD; ^4^ Underweight, weight for age z-score <−2SD; ^5^ Overweight, weight for age z-score > 2SD; ^6^ Wasted, weight for height/length z-score <−2 SD.

**Table 2 nutrients-13-02354-t002:** Quantities of complementary foods consumed per day (mean ± SD) * according to food category and age group.

Food Groups	Total (*N* = 119)			6–11.9 Months (*N* = 34)			12–17.9 Months (*N* = 41)			18–23.9 Months (*N* = 44)		
	Mean	SD	*N* (%)	Mean	SD	*N* (%)	Mean	SD	*N* (%)	Mean	SD	*N* (%)
Cereals and tubers (g)	179.5	85.4	118 (99.2)	123.9 ^a^	94.1	33 (97.1)	204.7 ^b^	78.8	41(100.0)	198.9 ^b^	63.4	44 (100.0)
Commercial Baby foods (g)	10.6	24.3	35 (29.4)	27.2 ^a^	35.8	21 (61.8)	4.2 ^b^	13.0	8 (19.5)	3.8 ^b^	13.5	6 (13.6)
Vegetables (g)	15.4	23.8	72 (60.5)	7.2 ^a^	15.6	14 (41.2)	16.5 ^a,b^	27.0	25 (61.0)	20.7 ^b^	24.6	33 (75.0)
Legumes (g)	1.7	10	19 (16.0)	0.2 ^a^	1.3	1 (2.9)	3.2 ^a,b^	16.7	7 (17.1)	1.3 ^b^	3.4	11 (25.0)
Fruit (g)	11.2	21.7	48 (40.3)	10.5 ^a^	21.9	13 (38.2)	7.8 ^a^	16.4	14 (34.1)	14.9 ^a^	25.5	21 (47.7)
Meat (g)	9.2	13.3	63 (52.9)	2.3 ^a^	4.1	9 (26.5)	9.5 ^b^	10.3	25 (61.0)	14.3 ^b^	17.7	29 (65.9)
Milk & dairy products (g)	17.5	51.7	31 (26.1)	0.1 ^a^	0.4	1 (2.9)	11.3 ^b^	28.9	10 (24.4)	36.7 ^b^	76.8	20 (45.5)
Egg (g)	4.1	10.7	33 (27.7)	0.0 ^a^	0.0	0 (0.0)	6.0 ^b^	10.1	16 (39.0)	5.5 ^b^	14.2	17 (38.6)
Fish & seafood products (g)	5	9.6	41 (34.5)	1.1 ^a^	3.5	4 (11.8)	4.1 ^a,b^	7.2	14 (34.1)	8.9 ^b^	13.0	23 (52.3)
Oils and fats	0.7	1.4	34 (28.6)	0.1 ^a^	0.4	1 (2.9)	0.6 ^b^	1.0	15 (36.6)	1.3 ^b^	2.0	18 (40.9)
Confectioneries, cakes (g)	4.2	10.6	40 (33.6)	0.4 ^a^	1.7	2 (5.9)	4.5 ^b^	12.7	14 (34.1)	6.8 ^b^	11.7	24 (54.5)
Western foods (g)	2.3	6.8	16 (13.4)	0.1 ^a^	0.9	1 (2.9)	2.6 ^a,b^	8.2	4 (9.8)	3.7 ^b^	7.7	11 (25.0)
Beverages (g)	4.1	25.4	7 (5.9)	0 ^a^	0	0(0)	2.2 ^a,b^	14.1	1 (2.4)	8.9 ^b^	39.3	6 (13.6)
Local cakes (“Kuih”) (g)	4.1	12.3	28 (23.5)	0.8 ^a^	2.8	5 (14.7)	6.0 ^a^	17.6	10 (24.4)	4.8 ^a^	10.5	13 (29.5)

* Determined using 2-day weighing records. Statistical differences among the age groups were tested using Kruskal–Wallis test and Dunn–Bonferroni post hoc; ^a,b^ values that do not share the same alphabets were significantly different from each other within the same row food group at *p* < 0.05. Cereal and tubers: rice, porridge, noodles, pasta, bread, biscuits, breakfast cereals, potato. Milk and dairy products: UHT milk, fresh milk, pasteurized milk, yogurt, cheese. Vegetables: vegetable soup, cooked vegetables, canned vegetables, commercial baby vegetables foods. Fruit: fresh fruit, fruit juice, commercial baby fruit foods. Commercial Baby foods: infant instant cereals, teething biscuits. Legumes: tofu, soybean milk. Confectionery: sweets, chocolate. Western foods: pizza, nuggets, frankfurters, hotdogs, sandwiches. Beverages: malted drinks, milkshake. Local cakes: “kuih”, rice flour-based, wheat flour-based, banana, cassava fritters, spring rolls.

**Table 3 nutrients-13-02354-t003:** Energy and nutrient intake from complementary foods per day (mean ± SD and median) according to age group.

Energy/Nutrients	Total (*N* = 119)			6–11.9 Months (*N* = 34)			12–17.9 Months (*N* = 41)			18–23.9 Months (*N* = 44)			*p*-Value ^1^
	Mean	SD	Median	Mean	SD	Median	Mean	SD	Median	Mean	SD	Median	
Energy													
Intake (kcal)	495.2	287.5	459.2	415.0 ^a^	308.2	309.5	510.0 ^b^	307.8	477.3	543.3 ^b^	241.1	493.9	0.037
% RNI	58.0	34.6	49.0	62.3	45.2	45.1	54.6	33.4	48.8	57.8	25.7	52.2	0.566
Protein													
Intake (g)	16.9	8.5	16.2	13.3 ^a^	8.6	11.9	17.3 ^b^	8.2	16.1	19.3 ^b^	8.0	17.4	0.005
% RNI	147.1	73.3	137.5	132.7	85.9	119.0	144.4	67.9	134.0	160.6	66.7	144.6	0.139
Vitamin A RE													
Intake (µg)	226.7	243.6	141.7	148.7 ^a^	158.9	96.3	236.3 ^a,b^	289.4	124.2	277.9 ^b^	240.7	203.3	0.006
% RNI	56.7	60.9	35.4	37.2 ^a^	39.7	24.1	59.1 ^a,b^	72.4	31.0	69.5 ^b^	60.2	50.8	0.006
Vitamin D													
Intake (µg)	1.0	1.4	0.4	1.1	1.7	0.3	0.7	0.9	0.4	1.1	1.4	0.6	0.608
% RNI	7.4	11.4	3.0	11.1	16.8	3.0	4.8	6.2	2.7	7.0	9.4	3.7	0.704
Thiamine													
Intake (mg)	0.28	0.27	0.21	0.27	0.20	0.20	0.26	0.31	0.18	0.32	0.29	0.26	0.128
% RNI	67.1	63.2	48.6	89.4 ^a^	66.0	65.8	51.9 ^b^	61.5	36.8	64.0 ^a^	58.8	52.9	0.003
Riboflavin													
Intake (mg)	0.41	0.30	0.36	0.38	0.26	0.33	0.38	0.30	0.33	0.47	0.33	0.40	0.314
% RNI	88.3	63.5	76.9	94.2	65.1	83.1	76.3	59.4	65.1	94.8	65.7	80.6	0.289
Niacin													
Intake (mg)	3.9	2.6	3.4	3.3 ^a^	2.2	2.7	3.8 ^b^	2.7	3.2	4.5 ^b^	2.8	4.2	0.020
% RNI	73.1	48.9	63.4	83.3	55.7	66.7	62.9	45.2	53.4	74.6	46.0	69.8	0.103
Folate													
Intake (µg DFE)	40.8	35.9	29.5	32.5 ^a^	34.2	18.45	38.0 ^a,b^	32.5	29.3	49.9 ^b^	39.0	36.2	0.008
% RNI	31.3	30.2	22.1	40.6	42.8	23.1	23.7	20.3	18.3	31.2	24.4	22.6	0.154
Vitamin C													
Intake (mg)	13.9	16.9	7.6	17.4	23.5	6.2	10.2	12.5	4.8	14.7	13.9	10.3	0.079
% RNI	46.4	56.4	25.2	58.0	78.4	20.8	33.9	41.7	16.2	48.9	46.2	34.4	0.079
Calcium													
Intake (mg)	143.3	129.3	91.2	157.4 ^a^	137.4	114.4	106.4 ^a^	106.3	81.5	166.7 ^b^	137.3	139.5	0.045
% RNI	31.3	36.8	19.2	60.5 ^a^	52.9	44.0	15.2 ^b^	15.2	11.6	23.8 ^b^	19.6	19.9	0.0001
Iron													
Intake (mg)	5.0	4.4	3.5	6.2	5.9	3.4	4.2	3.3	3.3	4.7	3.7	3.6	0.469
% RNI	72.8	59.9	51.4	68.8	65.3	38.2	69.8	54.5	54.6	78.6	61.4	59.7	0.125
Zinc													
Intake (mg)	1.4	0.9	1.28	1.1 ^a^	0.9	0.8	1.3 ^a^	0.8	1.1	1.8 ^b^	0.9	1.7	0.0001
% RNI	35.1	22.1	32.0	27.2 ^a^	23.1	20.3	32.0 ^a^	18.4	28.1	44.2 ^b^	21.8	42.0	0.0001

RNI: Recommended Nutrient Intake; ^1^ Statistical difference between three age groups were tested using Kruskal–Wallis test and Dunn–Bonferroni post hoc; ^a,b^ values that do not share the same alphabets were significantly different from each other within the same row nutrients at *p* < 0.05.

**Table 4 nutrients-13-02354-t004:** Intake of fats and fatty acids from complementary foods (mean ± SD and median) according to age group.

Fata and Fatty Acids	Total (*N* = 119)			6–11.9 Months (*N* = 34)			12–17.9 Months (*N* = 41)			18–23.9 Months (*N* = 44)		
	Mean	SD	Median	Mean	SD	Median	Mean	SD	Median	Mean	SD	Median
Total fat												
Intake (g)	8.7	6.5	6.9	3.5 ^a^	2.7	3.2	9.2 ^b^	6.3	7.7	12.1 ^b^	6.3	11.5
% RNI ^1^	28.6	20.6	24.3	14.1 ^a^	10.6	13.5	29.7 ^b^	20.3	25.7	38.8 ^b^	20.3	36.7

SFA intake (g)	3.0	2.5	2.4	1.1 ^a^	1.0	0.9	3.1 ^b^	2.2	2.9	4.4 ^b^	2.7	3.8
MUFA intake (g)	3.0	2.3	2.4	1.2 ^a^	0.9	1.1	3.3 ^b^	2.5	2.9	4.1 ^b^	2.2	3.9
PUFA intake (g)	1.6	1.3	1.4	0.9 ^a^	0.8	0.6	1.6 ^b^	1.3	1.3	2.2 ^b^	1.3	2.1
LA												
Intake (g)	1.3	1.2	1.1	0.7 ^a^	0.7	0.6	1.3 ^b^	1.3	1.1	1.8 ^b^	1.1	1.8
% AI ^2^	20.5	17.0	17.1	15.7 ^a^	15.2	13.0	18.2 ^a,b^	18.1	15.7	26.3 ^b^	16.0	25.0
ALA												
Intake (mg)	132	122	89	69 ^a^	73	47	142 ^b^	136	102	172 ^b^	122	129
% AI ^2^	20.0	17.8	14.0	13.7 ^a^	14.5	9.4	20.3 ^a,b^	19.5	14.6	24.6 ^b^	17.4	18.4
DHA + EPA intake (mg)	20	31	9	11 ^a^	18	5	23 ^b^	34	12	24 ^b^	34	12

RNI: Recommended Nutrient Intake; AI: Adequate Intake; SFA: Saturated fat; MUFA: Monounsaturated fat; PUFA: Polyunsaturated fat; LA: Linoleic fatty acid; ALA: Alpha linolenic acid; DHA: Docosahexaenoic fatty acid; EPA: Eicosapentaenoic fatty acid. ^1^ The absolute amounts of dietary fat recommended per day for the 6 to 11 months age group are calculated based on the current proposed Malaysian RNI for energy (630 kcal for boys aged 6 to 8 months; 720 kcal for boys aged 9 to 11 months; 570 kcal for girls aged 6 to 8 months; 660 kcal for girls aged 9 to 11 months) with 30–40% total energy intake as fat calories. For aged 12–23.9 months, dietary fat recommended per day are calculated based proposed Malaysian RNI for energy (980 kcal for boys and 900 kcal for girls) with 25–35% total energy intake as fat calories. ^2^ Adequate Intake (AI) for n-6 fatty acid for 0–6 months, 4.4 g/day; 7–12 months, 4.6 g/day; 1–3 years, 7 g/day. AI for *n*-3 fatty acids for 0–6 months, 500 mg/day; 7–12 months, 500 mg/day; 1–3 years, 700 mg/day (IOM, 2002). ^a,b^ values that do not share the same alphabets were significantly different from each other within the same row at *p* < 0.05.

**Table 5 nutrients-13-02354-t005:** Mean intake (g) per day and proportion of infants consuming milk-based formulas by age group (*n* = 95).

All Milk-Based Formulas (g) *	Total (*N* = 95)			6–11.9 Months (*N* = 27)			12–17.9 Months (*N* = 31)			18–23.9 Months (*N* = 37)		
	Mean	SD	N (%)	Mean	SD	N (%)	Mean	SD	N (%)	Mean	SD	N (%)
Cow milk-based formulas												
Formula C1 (g)	110.7	47.2	11 (11.6)	98.0	49.2	4 (14.8)	177.25	41.4	2 (6.5)	94.3	24.5	5 (13.5)
Formula C2 (g)	94.5	57.9	25 (26.3)	82.7	45.9	11 (40.7)	120.41	65.8	5 (16.5)	94.6	68.2	9 (24.3)
Formula C3 (g)	104.8	77.0	26 (27.4)	61.2	53.4	3 (11.1)	104.20	88.3	10 (32.3)	115.3	73.7	13 (35.1)
Formula C4 (g)	85.5	52.6	9 (9.5)	56.0	46.9	3 (11.1)	108.15	77.2	3 (9.7)	92.3	27.9	3 (8.1)
Formula C5 (g)	160.0	-	1 (1.1)	0	0	0 (0)	0	0	0 (0.0)	160.0	-	1 (2.7)
Formula C6 (g)	170.0	107.5	5 (5.3)	26.0	-	1 (3.7)	269.00	1.4	2 (6.5)	143.0	66.5	2 (5.4)
Formula C7 (g)	77.1	22.9	6 (6.3)	78.1	23.0	4 (14.8)	75.13	32.0	2 (6.5)	0	0	0 (0.0)
Formula C8 (g)	49.6	19.7	3 (3.2)	0	0	0 (0)	59.50	13.4	2 (6.5)	29.7	-	1 (2.7)
Formula C9 (g)	134.2	8.8	2 (2.1)	0	0	0 (0)	0	0	0 (0.0)	134.2	8.8	2 (5.4)
Formula C10 (g)	70.3	41.3	6 (6.3)	40.7	44.3	2 (7.4)	85.04	36.2	4 (12.9)	0	0	0 (0)
Formula C11 (g)	84	4	2 (2.1)	0	0	0 (0)	79	-	1 (3.2)	84	-	1 (2.7)
Formula C12 (g)	91.0	0.0	2 (2.1)	0	0	0 (0)	0	0	0 (0.0)	91.0	0.1	2 (5.4)
Soymilk-based formulas												
Formula S!	54.0	-	1(1.1)	54	-	1 (3.7)	0	0	0 (0.0)	0	0	0 (0.0)
Goat milk-based formulas												
Formula G1	47.0	-	1 (1.1)	0	0	0 (0.0)	0	0	0	47.0	-	1 (2.7)
Formula G2	15.0	-	1 (1.1)	0	0	0 (0.0)	15	-	1 (3.2)	0	0	0 (0.0)

* The commercial brands of the milk-based formulas are not disclosed for ethical reasons.

**Table 6 nutrients-13-02354-t006:** Energy and nutrient intake from milk-based formulas per day (mean ± SD and median) according to age group (*N* = 95).

	Total (*N* = 95)			6–11.9 Months (*N* = 27)			12–17.9 Months (*N* = 31)			18–23.9 Months (*N* = 37)			*p*
	Mean	SD	Median	Mean	SD	Median	Mean	SD	Median	Mean	SD	Median	
All formulas Intake (g)	103.7	60.6	95.7	78.8	42.5	78.6	115.0	76.3	112.0	112.3	52.7	101.5	0.051
Energy													
Intake (kcal)	480.6	276.1	443.8	382.0	206.4	388.8	534.3	349.2	500.6	507.4	237.1	471.0	0.136
% RNI	56.3	31.0	53.3	58.4	31.6	58.3	57.0	37.3	55.0	54.1	24.9	49.3	0.748
Protein													
Intake (g)	15.4	10.6	13.4	10.1 ^a^	6.4	9.6	17.1 ^a,b^	13.6	12.9	17.9 ^b^	8.8	17.2	0.002
% RNI	133.1	87.9	111.3	100.9 ^a^	64.1	95.8	142.2 ^a,b^	113.3	107.3	149.0 ^b^	73.3	142.9	0.031
Total fat													
Intake (µg)	19.5	11.4	17.3	18.0	10.2	18.7	21.5	14.3	19.0	18.9	9.2	17.1	0.796
% RNI	66.5	38.5	60.0	72.0	41.2	76.0	69.0	45.7	63.5	60.5	29.0	55.7	0.379
Vitamin A RE													
Intake (µg)	429.0	271.9	395.9	345.6	187.0	352.4	492.5	352.4	456.0	436.7	235.8	395.9	0.206
% RNI	107.3	68.0	99.0	86.4	46.8	88.1	123.1	88.1	114.0	109.2	58.9	99.0	0.206
Vitamin D													
Intake (µg)	7.8	5.3	6.5	5.2 ^a^	2.7	5.5	9.3 ^b^	6.7	8.4	8.3 ^b^	4.9	7.8	0.006
% RNI	56.8	35.7	54.0	51.9	27.5	55.0	62.3	44.8	56.0	55.7	32.7	52.0	0.722
Thiamine													
Intake (mg)	0.56	0.33	0.54	0.36 ^a^	0.19	0.35	0.61 ^b^	0.38	0.60	0.66 ^b^	0.31	0.64	0.0001
% RNI	125.9	66.6	120.8	121.4	64.5	118.0	121.9	75.7	120.0	132.6	61.1	128.4	0.774
Riboflavin													
Intake (mg)	0.86	0.52	0.81	0.66	0.38	0.68	0.96	0.66	0.88	0.91	0.46	0.83	0.131
% RNI	180.7	106.5	168.0	163.8	94.7	169.6	192.8	131.6	176.0	182.9	91.5	165.9	0.815
Niacin													
Intake (mg)	4.8	3.3	4.2	3.1 ^a^	1.7	3.1	4.7 ^a,b^	3.0	4.6	6.1 ^b^	3.9	5.4	0.002
% RNI	87.4	55.6	79.8	77.9	43.6	77.5	79.1	50.7	77.3	101.3	65.2	89.3	0.370
Folate													
Intake (µg DFE)	197.6	161.7	151.8	110.5 ^a^	65.4	103.6	209.0 ^a,b^	190.7	151.8	251.5 ^b^	161.5	226.4	0.0001
% RNI	143.1	102.1	121.9	138.1	81.8	129.5	130.6	119.2	94.9	157.2	100.9	141.5	0.209
Vitamin C													
Intake (mg)	76.1	58.9	67.6	57.5	32.1	65.2	89.5	72.9	69.2	78.3	58.9	61.6	0.248
% RNI	253.5	196.4	225.4	191.6	106.9	217.5	298.4	243.1	230.7	261.0	196.4	205.3	0.248
Calcium													
Intake (mg)	601.1	394.4	522.8	351.4 ^a^	217.1	304.2	687.0 ^b^	494.0	635.1	711.4 ^b^	321.1	662.2	0.0001
% RNI	110.0	67.7	98.4	135.1	83.5	117.0	98.1	70.6	90.7	101.6	45.9	94.6	0.131
Iron													
Intake (mg)	7.1	4.7	6.1	4.8 ^a^	2.9	4.7	8.2 ^b^	6.1	7.0	7.8 ^b^	3.8	7.1	0.006
% RNI	110.0	79.9	91.0	53.4 ^a^	32.1	51.9	135.9 ^b^	101.5	116.0	129.7 ^b^	63.0	118.3	0.0001
Zinc													
Intake (mg)	4.3	2.7	4.0	3.1 ^a^	1.7	3.1	4.8 ^a,b^	3.4	4.2	4.6 ^b^	2.4	4.6	0.029
% RNI	105.9	65.8	100.0	80.9	44.2	83.1	118.9	84.4	103.2	113.3	57.4	115.0	0.067

RNI: Recommended Nutrient Intake; DFE: Dietary folate equivalents. ^a,b^ values that do not share the same alphabets were significantly different from each other within the same row nutrients at *p* < 0.05.

**Table 7 nutrients-13-02354-t007:** Intake of fats and fatty acids from milk-based formulas per day (mean ± SD and median) according to age group (*N* = 95).

Fats and Fatty Acids	Total (*N* = 95)			6–11.9 Months (*N* = 27)			12–17.9 Months (*N* = 31)			18–23.9 Months (*N* = 37)		
	Mean	SD	Median	Mean	SD	Median	Mean	SD	Median	Mean	SD	Median
Total fat												
Intake (g)	19.5	11.4	17.3	18	10.2	18.7	21.5	14.3	19	18.9	9.2	17.1
% RNI ^1^	66.5	38.5	60.0	72.0	41.2	76.0	69.0	45.7	63.5	60.5	29.0	55.7

SFA intake (g)	8.2	5.6	7.1	7.3	4.7	6.6	9	7.2	7.2	8.3	4.7	7.2
MUFA intake (g)	7.0	4.7	6.5	6.7	4.1	6.4	7.8	5.7	7.1	6.7	4.2	5.7
PUFA intake (g)	3.2	2.1	3.3	3	1.7	3.3	3.6	2.2	3.4	3.1	2.2	2.7
LA												
Intake (g)	2.5	1.7	2.5	2.4	1.4	2.7	2.8	1.9	2.9	2.3	1.7	2.1
% AI ^2^	41.0	28.0	42.1	52.8 ^a^	31.4	58.7	39.9 ^a,b^	26.7	41.4	33.3 ^b^	24.1	30.0
ALA												
Intake (mg)	297	189	293	237	129	253	335	231	323	310	180	297
% AI ^2^	46.3	28.1	44.9	47.5	25.9	50.6	47.9	33.0	46.1	44.2	25.7	42.4
DHA + EPA intake (mg)	51	33	45	39	22	40	52	38	49	59	34	54

SFA: Saturated fat; MUFA: Monounsaturated fat; PUFA: Polyunsaturated fat; LA: Linoleic fatty acid; ALA: Alpha linolenic acid; DHA: Docosahexaenoic fatty acid; EPA: Eicosapentaenoic fatty acid. ^1^ Recommended Nutrient Intake (RNI) for the absolute amounts of dietary fat recommended per day for the 6 to 11 months age group are calculated based on the current Malaysian RNI for energy (630 kcal for boys aged 6 to 8 months; 720 kcal for boys aged 9 to 11 months; 570 kcal for girls aged 6 to 8 months; 660 kcal for girls aged 9 to 11 months) with 30–40% total energy intake as fat calories. For aged 12–23.9 months, dietary fat recommended per day are calculated based Malaysian RNI for energy (980 kcal for boys and 900 kcal for girls) with 25–35% total energy intake as fat calories. ^2^ Adequate Intake (AI) for n-6 fatty acid for 0–6 months, 4.4 g/day; 7–12 months, 4.6 g/day; 1–3 years, 7 g/day. AI for n-3 fatty acids for 0–6 months, 500 mg/day; 7–12 months, 500 mg/day; 1–3 years, 700 mg/day (IOM, 2002). Statistical difference between three age groups were tested using Kruskal–Wallis test and Dunn–Bonferroni post hoc; ^a,b^ values that do not share the same alphabets were significantly different from each other within the same row at *p* < 0.05.

**Table 8 nutrients-13-02354-t008:** Comparing intake of energy and nutrients from complementary foods and milk-based formulas in terms of contributions to recommended intake (RNI) by age group (*N* = 95).

Nutrients	Total				6–11.9 Months				12–17.9 Months				18–23.9 Months			
	Complementary Foods (*N* = 119)		Formula Milk (*N* = 95)		Complementary Foods (*N* = 34)		Formula Milk (*N* = 27)		Complementary Foods (*N* = 41)		Formula Milk (*N* = 31)		Complementary Foods (*N* = 44)		Formula Milk (*N* = 37)	
	Mean	Median	Mean	Median	Mean	Median	Mean	Median	Mean	Median	Mean	Median	Mean	Median	Mean	Median
Energy																
% RNI	58 ± 35	49	56 ± 31	53	62 ± 45	45	58 ± 32	58	55 ± 33	49	57 ± 37	55	58 ± 26	52	54 ± 25	49
Protein																
% RNI	147 ± 73	138	133 ± 88	111	133 ± 86	119	101 ± 64	96	144 ± 68	134	142 ± 113	107	161 ± 67	145	149 ± 73	143
Total fat																
% RNI	29 ± 21	24	67 ± 39	60	14 ± 11	14	72 ± 41	76	30 ± 20	26	69 ± 46	63	39 ± 20	20	61 ± 29	56
Vit A																
% RNI	57 ± 61	35	107 ± 68	99	37 ± 40	24	86 ± 47	88	59 ± 72	31	123 ± 88	114	70 ± 60	51	109 ± 59	99
Iron																
% RNI	73 ± 60	51	110 ± 80	91	69 ± 65	38	53 ± 32	52	70 ± 55	55	136 ± 102	116	79 ± 61	60	130 ± 63	118
Zinc																
% RNI	35 ± 22	32	106 ± 66	100	27 ± 23	20	81 ± 44	83	32 ± 18	28	119 ± 84	103	44 ± 22	42	113 ± 57	115
Calcium																
% RNI	31 ± 37	19	110 ± 68	98	61 ± 53	44	135 ± 84	117	15 ± 15	12	98 ± 71	91	24 ± 20	20	102 ± 46	95
Folate																
% RNI	31 ± 30	22	143 ± 102	122	41 ± 43	23	138 ± 82	130	24 ± 20	18	131 ± 119	95	31 ± 24	23	157 ± 101	142

## Data Availability

The data presented in this study are available within the article, except for Table 5, for which the commercial brands of the milk-based formulas are not disclosed for ethical reasons.

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
