# Peer review of "Complementary Foods and Milk-Based Formulas Provide Excess Protein but Suboptimal Key Micronutrients and Essential Fatty Acids in the Intakes of Infants and Toddlers from Urban Settings in Malaysia"

_nutrients, 2021, doi:10.3390/nu13072354_

Round 1

Reviewer 1 Report

Thank you for taking the time to take my comments into consideration and for refining this manuscript. It reads well. 

Author Response

Thank  you very much for your positive comments.

Reviewer 2 Report

Manuscript nutrients-1266807

Review

Scope of the journal: I think the paper is in the scope of Nutritiens

Importance: This study is of great importance since it highlights nutritional risks in small infants.

The study design fine. It is an observational study taken place in a society where the researcher needs to interact with infants and families, in suburbs of two large cities in Malaysia. Quite demaning task!

Soundness of conclusions and interpretation are in perfect order.

The relevance of the discussion is great and very relevant.

The paper is clear written.

Page 2, line 75,  …..snow-ball effect…please use another word. It is har to understand what the authors mean in this context.

References up to date and relevant.

Author Response

Thank you very much for your most encouraging comments.

I have taken heed of your comment to replace the term "Using the snow-ball approach in going from one center to the next,        on page 2 line 75.  

The term is replaced with "Using a convenient sampling approach in going from one center to the next,     

This manuscript is a resubmission of an earlier submission. The following is a list of the peer review reports and author responses from that submission.

Round 1

Reviewer 1 Report

General

  • This is a well written, scientifically strong manuscript that shares important data from Malawi albeit a small sample and tries to support this with data in other sites and compared to standards.
  • The manuscript would benefit from a close review of the grammar and text.

Introduction

  • The first sentence should be paraphrased rather than a direct quote from an article. It would be helpful to clarify the data that is being highlighted from the Lancet report and which report this is before mentioning the Lancet report in the next sentence. I would suggest revisiting and determining what is important in this sentence to support the introduction. For lines 33-35, I suggest moving them to the end of the introduction before presenting the objective of this paper.
  • Suggest starting with global burden that is mentioned on lines 36+ before delving into what is happening in Malaysia in the first few lines.
  • Line 38 add “an” before “adverse.”
  • Line 60 add “an” before “assessment.”
  • Line 67 and 68: Change the current report to this manuscript/paper/study
  • Line 67 change “as” to “by” before “Khor...”
  • Line 68 “determine” instead of “determined.”

Materials and methods

  • Please mention what the study design was.
  • Row 83 extra space before “meanwhile.”
  • Row 85 period needed.
  • Row 88 delete “a.”
  • Row 87-92: how did you choose the 40 infants in each group? Was it random? Was it the first 40? Please explain.
  • Row 93 “outweighing” should be 2 words.
  • Row 94 what center? How many centers?
  • Row 98 and 99: were the foods and beverages prepared the same way for all infants?
  • Row 116 extra space before “besides.”
  • Row 122 spell out FAO
  • Row 141 “5years” should be 2 words.
  • Row 146 rather than ”n (%)” mention “frequencies.”

Results

  • Row 169 change spelling to “practicing”; and “mode” not needed
  • Row 177 Was 1 subject stunted and wasted or 6? Please clarify.
  • Table 1, I think the “normal” rows are not needed.
  • Row 200 spell out UHT.
  • Row 324 change “as” to “a.”
  • All tables, in title change n to N and make months plural.

Discussion

  • First paragraph “energy” is used a lot. Might want to change some of the wording to avoid repetition.

Reviewer 2 Report

This study was to measure the intakes of complementary foods (CFs) and milk-based formulas (MFs) by subjects aged 6-23.9 months from urban daycare centers. But some questions are needed to clarify, such as

  1. The study should discuss the representation of the study population. Total 300 infants enrolled from a day care center, but only 119 infants were incomplete measured as one dietary intake. How to select the daycare center?

How to estimate dietary intake accurately using a two-day weight food record? Do you perform the reproducible and consistent test between student research assistants?  

  1. All measurements for dietary intake should have good precision and accuracy, so it needs to clearly describe the SOP or QA/QC. Why three age strata were classified in the article? Do you have any meaning in the three age groups based on the guidelines in Malaysia or USDA?
  2. Although Many Tables were provided in the article, they should focus on the objective of this study. Does the article have any hypotheses? It just describes the raw data in three age groups rather than examining the hypothesis.
  3. Do you perform the multivariate analysis to find the risk factors on dietary behaviors? Based on the RNI, there was a high variation in various nutrients in Table 6. Can you explain the reasons or obtain from measurement error?   
  4. Because mothers from Malaysia have multi-racial people, which have different foods and milks-based formulas, it is crucial to classify different racial backgrounds of mothers or families, such as Chinese or Malaya. Notably, mother mild did not include in the study even the study focus on foods and mil-based formula only. It may be completely to meet RNIs for nutrients.
  5. The article should cite the previous studies regarding the national surveys or the study with the same age group. The article mentions is the first study to assess the dietary intake of infant and toddlers in Malaysia however, NCI has set the guideline to recommend for the population.
  6. Does the article test the association between Anthropometric measurements and dietary behavior?
  7. The limitations should be added to the questions from presentative, measurement method, no included other risk factors, etc.